# Adducts of Carbon Black with a Biosourced Janus Molecule for Elastomeric Composites with Lower Dissipation of Energy

**DOI:** 10.3390/polym15143120

**Published:** 2023-07-22

**Authors:** Federica Magaletti, Fatima Margani, Alessandro Monti, Roshanak Dezyani, Gea Prioglio, Ulrich Giese, Vincenzina Barbera, Maurizio Stefano Galimberti

**Affiliations:** 1Department of Chemistry, Materials and Chemical Engineering “G. Natta”, Politecnico di Milano, Via Mancinelli 7, 20131 Milano, Italy; federica.magaletti@polimi.it (F.M.); fatima.margani@polimi.it (F.M.); alessandro16.monti@mail.polimi.it (A.M.); roshanak.dezyani@mail.polimi.it (R.D.); gea.prioglio@polimi.it (G.P.); 2Deutsches Institut für Kautschuktechnologie e. V., Eupener Straße 33, 30519 Hannover, Germany; ulrich.giese@dikautschuk.de

**Keywords:** carbon black, elastomers, nanocomposites, rubber, functionalization

## Abstract

Elastomer composites with low hysteresis are of great importance for sustainable development, as they find application in billions of tires. For these composites, a filler such as silica, able to establish a chemical bond with the elastomer chains, is used, in spite of its technical drawbacks. In this work, a furnace carbon black (CB) functionalized with polar groups was used in replacement of silica, obtaining lower hysteresis. CBN326 was functionalized with 2-(2,5-dimethyl-1*H*-pyrrol-1-yl)-1,3-propanediol (serinol pyrrole, SP), and samples of CB/SP adducts were prepared with different SP content, ranging from four to seven parts per hundred carbon (phc). The entire process, from the synthesis of SP to the preparation of the CB/SP adduct, was characterized by a yield close to 80%. The functionalization did not alter the bulk structure of CB. Composites were prepared, based on diene rubbers—poly(1,4-*cis*-isoprene) from *Hevea Brasiliensis* and poly(1,4-*cis*-butadiene) in a first study and synthetic poly(1,4-*cis*-isoprene) in a second study—and were crosslinked with a sulfur-based system. A CB/silica hybrid filler system (30/35 parts) was used and the partial replacement (66% by volume) of silica with CB/SP was performed. The composites with CB/SP exhibited more efficient crosslinking, a lower Payne effect and higher dynamic rigidity, for all the SP content, with the effect of the functionalized CB consistently increasing the amount of SP. Lower hysteresis was obtained for the composites with CB/SP. A CB/SP adduct with approximately 6 phc of SP, used in place of silica, resulted in a reduction in ΔG′/G′ of more than 10% and an increase in E’ at 70 °C and in σ_300_ in tensile measurements of about 35% and 30%, respectively. The results of this work increase the degrees of freedom for preparing elastomer composites with low hysteresis, allowing for the use of either silica or CB as filler, with a potentially great impact on an industrial scale.

## 1. Introduction

Global greenhouse emissions, measured in carbon dioxide equivalents over a 100-year timescale, were 54.59 billion tons in 2021 [1]. The United Nations (UN) has recognized the importance of sustainable transport since the 1992 Earth Summit [2,3]. In fact, the energy sector contributes more than 70% to global greenhouse emissions, with road transport accounting for about 12% [4]. According to the UN’s mobility report, global freight volumes and annual passenger traffic are projected to grow by 50% and 70%, respectively, compared to 2015, with 2.4 billion cars on the road. In this context, tires play a major role, with 2.7 billion units forecast to be on the market by 2025 [5,6], contributing about 20% to the global warming potential, 90% of which is derived from the use of cars. So-called rolling resistance (RR), i.e., “the energy consumed per unit distance of travel as a tire rolls under load” [7,8,9] is mostly responsible for the dissipation of energy and, hence, for the environmental impact of a tire during its use. RR is influenced by the hysteresis of an elastomeric composite, which in turn is mainly due to the so-called filler networking phenomenon, that is, the reversible interactions of the reinforcing filler particles [10,11,12,13].

Furnace carbon black (CB) [14,15,16,17,18,19] and precipitated silica [20,21] are the traditional reinforcing fillers in tire compounds: they possess a high surface area and therefore high interfacial area with the rubber matrix and are nanostructured, which means their aggregates have voids able to occlude polymer chains, thereby promoting mechanical reinforcement. Silica is the preferred filler for reducing the hysteresis of a rubber compound: the silanols on the silica surface react with so-called coupling agents, which enable compatibility with the silica and establish chemical bonds with the polymer chains [20,21], reducing if not preventing the filler networking phenomenon. In order to reduce the hysteresis of elastomeric composites for a dynamic-mechanical application such as the one in tire, the filler–polymer chemical bond is a key feature, alongside the appropriate selection of materials. The preferred coupling agents are silanes containing sulfur atoms [22,23], such as bis(triethoxysilylpropyl)tetrasulfide (TESPT) [22]. Silica-based technology is well established and the use of silica in tire compounds is steadily increasing. However, silica presents remarkable drawbacks. It promotes the increase of compound viscosity, and this leads to worse processability and to a shorter storage time of the composites, which makes it necessary to change the planning of the production of the compounds as well as the procedures for storing and moving them, with a clear impact on logistics. Particular mixing equipment has to be used, which is more energy consuming and more expensive. Silica is corrosive and abrasive, and the silane causes increased adhesiveness to the metal parts of the mixing machines. Hence, the metal surfaces have to be treated with special substances, which requires a revision of maintenance procedures. Moreover, compounds based on silica suffer from a lack of electrical conductivity. All these technical problems are particularly relevant on an industrial scale. It would be highly desirable to use, as a reinforcing filler for an elastomeric composite with low hysteresis, a CB with the properties of traditional CB grades but with chemical reactivity with rubber chains. This objective can be achieved through introducing functional groups on the CB surface.

Some of the authors developed a functionalization method [24,25,26,27,28,29] which uses biosourced molecules, is performed in the absence of solvents and catalyts and is characterized by high yield (also higher than 95%) and carbon efficiency [30,31]. A solvent (acetone) was used only to allow for the easy mixing of CB and the pyrrole compound in the lab. This method is based on pyrrole compounds, with their general chemical structure shown in Figure 1.

The functionalization of CB [24,25], a high-surface-area graphite (HSAG) [26,27] and carbon nanotubes [28] was performed through simply mixing the carbon materials and a pyrrole compound (PyC), giving either thermal or mechanical energy. A domino reaction occurs, in which the steps are the carbocatalyzed oxidation of the pyrrole compound in the benzylic position and then the cycloaddition reaction with the carbon substrate [29]. It is worth highlighting the versatility of this functionalization reaction, which allows us to attach nearly any type of R group. A large part of the research was performed with 2-(2,5-dimethyl-1*H*-pyrrol-1-yl)-1,3-propanediol (serinol pyrrole, SP), whose chemical structure is in Figure 1b. SP was prepared from the Paal Knorr reaction of a biosourced molecule, a glycerol derivative such as 2-amino-1,3-propandiol (known as serinol), with 2,5-hexanedione, which could be prepared from dimethylfuran [32,33,34,35]. SP is a Janus molecule [27,36]: the pyrrole ring gives rise to a covalent bond with the carbon substrate and the serinol moiety changes the solubility parameter of the carbon allotrope [27,28] and promotes its chemical reactivity.

The CB/SP adduct appears to be an ideal filler for preparing rubber composites with low dissipation of energy, avoiding or substantially reducing the use of silica. The OH groups brought by SP on the CB surface can promote the interaction of CB with silica and with the coupling agent TESPT, and hence with the elastomer chains. In this work, CB/SP was used in replacement of a major amount of silica, in elastomeric composites based on diene rubbers, with a CB/silica hybrid filler system. The objective was to at least reproduce the properties of the silica-based compound, above all the low hysteresis. For the preparation of the CB/SP adducts, CB N326 was used. According to the ASTM-D1765 standard classification system for carbon blacks used in rubber products, this grade of CB has a surface area of about 80 m^2^/g, remarkably lower than the one of the sp^2^ carbon allotropes used by some of the authors for the functionalization with SP: 300 m^2^/g for HSAG [26] and 250 m^2^/g for CNT [28]. Hence, this work also had the objective to investigate the robustness of the methodology based on pyrrole compounds for the functionalization of CB. This technology appears of interest for its development to the industrial scale. A major player in the tire field has reported the use of a pyrrole compound, and in particular, of serinol pyrrole, for the industrial development [37].

The CB/SP adducts were characterized by means of thermogravimetric analysis (TGA) and X-ray diffraction.

In a first study, the CB/SP adduct was used for the partial replacement of silica in a composite based on poly(1,4-*cis*-isoprene) from *Hevea Brasiliensis* (natural rubber, NR) and poly(1,4-*cis*-butadiene) with high 1,4-*cis* content. This type of composite is typically used in an important tire compound such as the sidewall. In a second study, the CB/SP adduct was analogously used in partial replacement of silica, with poly(1,4-*cis*-isoprene) as the only rubber. Internal tire compounds, for carcass and belt, are typically based on this type of rubber. In most cases, NR is used. However, to ensure the reproducibility of the compounds’ properties, the synthetic poly(1,4-*cis*-isoprene) (IR) from Ziegler-Natta catalysis is sometimes used in place of NR. In this study, the choice of IR in place of NR was aimed at avoiding the interaction of the polar CB/SP with the polar groups which are present at the NR chain ends, proteins and phospholipids. The compounds were crosslinked with a sulfur-based system and were characterized by means of dynamic-mechanical tests, both in the shear and in the axial mode, and tensile measurements.

## 2. Materials and Methods

### 2.1. Materials

#### 2.1.1. Chemicals

All reagents and solvents were purchased and used without further purification: 2-amino-1,3-propanediol, 2,5-hexanedione, acetone from Sigma-Aldrich; TESPD 3,3′-bis(triethoxysilylpropyl)disulfide from Flexys, Ann Arbor, MI, USA.

The following chemicals have been used for the preparation of the elastomeric composites discussed in this paper: X50S (50% carbon black, 50% silane, Degussa, Milan, Italy), ZnO (Zincol Ossidi, Bellusco, MB, Italy), Stearic acid (Sogis, Milan, Italy), 1,3-dimethyl butyl)-*N*′-Phenyl-p-phenylenediamine (6PPD from Eastman, Kingsport, TN, USA), Sulfur (S from Solfotecnica, Cotignola, Italy), *N*-tert-butyl-2-benzothiazyl sulfenamide (TBBS from Lanxess Chemical, Shangai, China), *N*-(Cyclohexylthio)phthalimide (PVI, Brenntag, S.p.A., Milan, Italy).

#### 2.1.2. Elastomers

Poly(1,4-*cis*-isoprene) from Hevea brasiliensis (NR) (EQR-E.Q. Rubber, BR-THAI, Eastern GR. Thailand—Chonburi) had the trade name SIR20 and 73 Mooney Units (MU) as Mooney viscosity (ML(1 + 4)100 °C). Synthetic poly (1,4-butadiene) (BR) was Neocis BR 40 from Versalis, with a 43 Mooney Viscosity (ML(1 + 4)100 °C). Synthetic poly-1,4-*cis*-isoprene (IR) was from Nizhnekamskneftechim Export, with the trade name SKI3 and 70 Mooney Units (MU) as Mooney viscosity (ML (1 + 4) 100 °C).

#### 2.1.3. Fillers

Carbon black N326 was kindly provided by Birla Carbon (Atlanta, GA, USA). Data from the technical data sheet are as follows. Oil absorption number (OAN): 72 mL/1000, nitrogen specific surface area (NSA): 78 m^2^/g, statistical thickness surface area (STSA): 78 m^2^/g.

Silica ZEOSIL 1165 MP were white micropearls from Solvay (Brussels, Belgium). Data in the technical data sheet are as follows. Specific surface area: 140–180 m^2^/g, loss on drying (2 h @ 105 °C) ≤8.0%, soluble salts (as Na_2_SO_4_) ≤2.0%. In this work, the surface area was determined using the BET method. Samples were evacuated at 200 °C for 2 h and N_2_ adsorption isotherms were recorded at 77 K in a liquid nitrogen bath using a MICROACTIVE TRISTAR ^®^ II PLUS apparatus. The specific surface area (SSA) was found to be 160 m^2^/g.

### 2.2. Preparation of SP and CB/SP Adducts

#### 2.2.1. Synthesis of 2-2,5-Dimethyl-1*H*-pyrrol-1-yl-1,3-propanediol (SP)

In a round bottom flask, equipped with a condenser and magnetic stirrer, 16.23 g of serinol (0.1781 mol) was suspended in 21 mL of 2,5-hexanedione (0.1780 mol, 1 eq), the temperature was raised to 150 °C and stirring was performed for 2 h. The condenser was then removed, and the reaction mixture was stirred for a further 30 min. The water evaporated during the synthesis. The product was collected as a brown, viscous liquid without any further purification (27.73 g, 0.1639 mol). The yield was calculated using the following expression: 100 × (weighed mass of SP)/(theoretical mass of SP). It was calculated to be about 92%. NMR spectra are reported in the Appendix A.

^1^H NMR (CDCl_3_, 400 MHz); δ (ppm) = 2.27 (s, 6H), 3.99 (m, 4H); 4.42 (quintet, 1H); 5.79 (s, 2H). ^13^C NMR (DMSO-*d*_6_, 100 MHz); δ (ppm) = 127.7, 105.9, 43.72, 71.6, 61.2, 13.9.

#### 2.2.2. Preparation of Adducts of CB N326 with SP

Pristine CB N326 (1.0 g) and SP dissolved in acetone were poured in a 50 mL round flask, with a total amount of acetone suitable to completely cover the powder. The mixture was sonicated for 15 min. The solvent was then removed under reduced pressure and the resulting dry powder was heated through immersing the flask in a silicon bath.

The dry powder was then heated at different temperatures in order to obtain three different adducts, in particular: (i) at 150 °C functionalization temperature with SP at 10 phc (CB/SP-6); (ii) at 120 °C functionalization temperature with SP at 10 phc (CB/SP 5); (iii) at 120 °C functionalization temperature with SP at 5 phc (CB/SP-4).

After the reaction, the adduct CB-SP was cooled to room temperature and then washed with a Soxhlet extractor overnight, using acetone as the solvent. The wet powder was dried in the oven at 80 °C for 1 h. The dry powder and the SP possibly extracted were not weighed. The functionalization yield, reported in the following, was thus estimated by means of TGA measurements.

### 2.3. Preparation of Elastomer Composites

#### 2.3.1. Elastomer Composite with CB/SP-6

The recipe is in Table 1. The reference composite, without CB/SP, is indicated as the “silica” composite. The CB/silica hybrid filler system, in the (30/35 parts) ratio, was adopted in previous works (results not reported) and appeared to be suitable, in the present study, to investigate the behavior of CB/SP. The processing procedure is shown in Figure 2.

#### 2.3.2. Elastomer Composite with CB/SP-4 and CB/SP-5

The recipe is in Table 2. The reference composite, without CB/SP, is indicated as the “silica” composite. The processing procedure is shown in Figure 2.

### 2.4. Characterization Techniques

#### 2.4.1. Thermogravimetry Analysis (TGA) of CB/SP Adducts

TGA analysis were performed with a TGA TA instrument Q500 (TA instrument, New castle, DE) according to the following method: heating under a nitrogen blanket from 30 °C to 300 °C at 10 °C/min, isothermal step at 300 °C for 15 min, heating to 550 °C at 20 °C/min, isothermal step for 15 min, heating to 900 °C at 10 °C/min, isothermal step for 3 min, shift from nitrogen to air, final isothermal step for 30 min.

In this work, the amount of SP in the CB/SP adduct is expressed in parts per hundred carbon (phc), which were estimated using Equation (1), where (weight loss)_150–900°C_ = x.
(1)$$SP\;(phc)=100\cdot \frac{x_{\;}}{\left(100-x\right)}$$

The yield of the functionalization process, from the weight of reactants to the estimation of the mass loss of CB/SP adduct via TGA, was calculated by means of Equation (2).
(2)$$Process\;yield\;(\%)=100\;\cdot \frac{SP\;mass\;\%\;in\;{\left(CB/SP\;adducts\right)}_{after\;washing}}{SP\;mass\;\%\;in\;CB-{SP}_{reactants\;mixture}}$$

#### 2.4.2. Wide-Angle X-ray Diffraction

With a Bruker D8 Advance automatic diffractometer (Bruker Italia SRL, Milan, Italy) and nickel-filtered Cu-Kα radiation, wide-angle X-ray diffraction patterns were carried out in reflection. The range of 4.7° to 90° was used to record the patterns because these angles were the 2θ peak diffraction angles.

Spectra were developed using Origin Pro 2018.

From the Bragg law (Equation (3)), the distance between crystallographic planes was estimated.
(3)$$d_{hkl}=\frac{n\cdot \lambda }{2\cdot \sin\theta_{hkl}}$$
where *n* is an integer number, *λ* is the wavelength of the irradiating beam and *θ_hkl_* is the diffraction angle.

The Scherrer equation (Equation (4)) was used to calculate the *D_hkl_* crystallite dimensions using the chosen planes.
(4)$$D_{hkl}=\frac{k\cdot ʎ}{\beta_{hkl}\cdot \cos\theta_{hkl}}$$
where *β_hkl_* is the width at half height, *θ_hkl_* is the diffraction angle, *K* is the Scherrer constant and *λ* is the wavelength of the irradiating beam.

As shown in the following equation (Equation (5)), the number of layers was simply calculated through dividing the crystallite size of the (002) plane, which corresponds to the perpendicular dimension of the multi-layered material, by the interlayer distance.
(5)$$number\;of\;layers=\frac{D_{hkl}}{d_{hkl}}$$

#### 2.4.3. Crosslinking

It was performed in a rubber process analyzer (Monsanto R.P.A. 2000, Alpha Technologies Hudson, Ohio, USA). An amount of 5 g of rubber composite was weighed and put in the rheometer. Measurements were carried out at a frequency of 1.7 Hz and an oscillation angle of 6.98%. The sample, loaded at 50 °C, underwent a first strain sweep (0.2–25% strain) to cancel the thermo-mechanical history of the rubber composite; it was then maintained at 50 °C for 10 min and then underwent another strain sweep at 50 °C to measure the dynamic-mechanical properties at low deformations of the uncured sample. A crosslinking reaction was then carried out at 170 °C for 10 min. A torque–time curve was obtained. The minimum achievable torque (M_L_), the maximum achievable torque (M_H_), the time required to have a torque equal to M_L_ + 1 (t_S1_) and the time required to reach 90% of the maximum torque (t_90_) were measured.

#### 2.4.4. Dynamic-Mechanical Analysis in the Shear Mode Strain Sweep Test

The shear dynamic-mechanical characteristics of the rubber compounds were evaluated through performing strain sweep tests in a rubber process analyzer (Monsanto R.P.A. 2000, Alpha Technologies Hudson, Ohio, USA). As reported in Section 2.4.3, the crude sample was subjected to a first strain sweep, was then held at 50 °C for ten minutes, followed by another strain sweep at 50 °C. Data from the second strain sweep were collected and are reported in the text below to discuss the behavior of uncured samples. The crosslinking was then carried out, as reported in Section 2.4.3, and the shear dynamic-mechanical properties of the cured samples were then assessed after 20 min at 50 °C using a 0.2–25% strain sweep at a frequency of 1 Hz. Shear storage and loss moduli (G′, G″), and subsequently Tan δ, were measured characteristics.

#### 2.4.5. Dynamic-Mechanical Analysis in the Axial Mode

The elastomeric compound was rolled up to obtain a long cylinder. This cylinder was then cut into smaller cylinders and vulcanized (at 170 °C for 10 min) to produce cylindrical test pieces with dimensions of 25 mm in length and 12 mm in diameter. An Instron dynamic device (Instron, Buckinghamshire, UK) in traction–compression mode was employed to perform dynamic mechanical measurements and was maintained at the predetermined temperatures (10, 23 and 70 °C) throughout the entire experiment. The cylinder was preloaded to a 25% longitudinal deformation with respect to the original length. The compression was subjected to a dynamic sinusoidal strain in compression with an amplitude of around 3.5% regarding the length under pre-load, at a frequency of 100 Hz. This generated the values of dynamic storage modulus (E′) and loss modulus (E″), as well as loss factor (tan δ), calculated as the ratio between E″ and E′.

#### 2.4.6. Tensile Test

Standard dumbbells made from 10 cm by 10 cm by 1 mm vulcanized compound plates were used to perform tensile tests at room temperature with a Zwick Roell Z010 and an optical extensometer. Measurements were performed at 1 mm/min. Stresses at different elongations (respectively σ_50_, σ_100_, σ_300_), stress at break (σ_B_), elongation at break (ε_B_) and the energy required to break were measured according to Standard ISO [38].

#### 2.4.7. Electrical Resistance

Keysight Technologies 34,450 A 5 ½ Digital Multimeter (Keysight Technologies SRL, Milan, Italy) was used to measure the electrical resistance of rubber composites. The measurements were taken with a hand-applied four-point probe (FPP) with four distinct gold-coated points. (Each tip has an area of 18 mm^2^). Only two of the FPP’s points were used as crocodile clips for volumetric measurement. The samples are 3.5 cm diameter cured rubber compound disks with a thickness of 3 mm. For each composite, five measurements were taken.

#### 2.4.8. Headspace Analyses

Headspace analyses were carried out using an Agilent HS-Autosampler 7697A, an Agilent GC 6890 N and an Agilent MS 5975C.

## 3. Results and Discussion

### 3.1. Preparation and Characterization of CB/SP Adducts

The synthetic pathway for the preparation of the CB/SP started from the synthesis of SP, which was carried out as described elsewhere [28]. Quantitative yield [39] was achieved (96%), with water as the only co-product and thus with very high carbon efficiency [30,31], i.e., with a high amount of the reagents’ carbon atoms present in the final product.

For the preparation of the CB/SP adducts, CB N326 was used. As mentioned in the Introduction, this grade of carbon black has a surface area of about 80 m^2^/g, remarkably lower than the one of the sp^2^ carbon allotropes used by some of the authors for functionalization with SP.

The procedure for the preparation of the CB/SP adduct is in Figure 3.

Details are in the experimental section. In brief, CB and SP were mixed in acetone, and sonication was then performed to prepare a homogeneous dispersion. After the removal of the solvent, the temperature was increased, either to 120 °C or to 150 °C, and the reaction was carried out for 4 h. The adducts were then washed with acetone, to completely remove the unreacted SP. In this procedure, already used for preparing adducts with HSAG [27] and CNT [28], a solvent such as acetone was used, to allow for easy mixing at the lab scale. For the scale-up of the reaction, an organic solvent for the preparation of the physical mixture could be avoided, for example, through using an apparatus such as a spray dryer [40].

Adducts with different amounts of SP were prepared through adopting different reaction temperatures and different amounts of SP in the reaction mixture. When the reaction was performed at 150 °C, 10 parts of SP per hundred parts of CB (phc) was used. In the case of the reaction carried out at 120 °C, 5 and 10 phc of SP were used. The amount of SP in the adducts was estimated by means of TGA analysis. Data of mass losses are in Table 3. The thermographs are in Appendix A. Three main steps can be seen in the decomposition profile: at a temperature below 150 °C, between 150 °C and 800 °C and above 900 °C. Low-molar-mass substances, such as adsorbed water or the acetone used for washing, could be responsible for the mass loss at T < 150 °C. The decomposition of PyC and the alkenylic defects of CB can account for the mass loss between 150 °C and 900 °C. Residues at T > 900 °C were not observed or were present in a very low amount (about 0.5%).

The content of SP in the adducts (in phc) was estimated on the basis of the mass loss between 150 °C and 900 °C, which, as mentioned above, could be also due to the defects of CB, which, however, are supposed to react with SP to form the adducts. The estimation of the SP content was made by means of Equation (1) (see Section 2.4.1) and is shown, with phc as the measure unit, in Table 3. In the following, the adducts are named with a round figure indicating the level of functionalization. The yield of the whole functionalization process was estimated by means of Equation (2) (see Section 2.4.1). Through performing the functionalization with a higher amount of SP (10 phc) at 150 °C, a yield of about 60% (at 150 °C) or about 50% (at 120 °C) was achieved, lower than the ones reported in the case of HSAG [26,27] and CNT [28], sp^2^ carbon allotropes with remarkably higher surface area. Functionalization carried out with a lower amount of SP (5 phc) led to about 80% functionalization. The amount of SP in the reaction mixture seems to play a major role with respect to the functionalization temperature. The moderate surface area of CBN326 could account for these findings. In consideration of the yield for the SP synthesis, in the case of CB/SP-4, a yield close to 80% was obtained for the global process for preparing the CB/SP adduct.

X-ray analysis was performed on CB and on the adducts CB/SP-4 and CB/SP-5. CB is made up of poorly organized areas (disordered and amorphous carbons) and, mostly, of aggregates made by a few graphene layers, assembled in spherical particles. The patterns are in Figure 4, and the parameters indicating the size of crystallites in the direction orthogonal and parallel to the structural layers, the interlayer distance and the anisotropic index are in Appendix A.

The diffraction peaks were observed at the following 2θ values: 24.46° (002) and 43.99° (100) for CB/SP-4 and at 24.41° (002) and 43.79° (100) for CB/SP-5. The values reported in Appendix A indicate that the functionalization did not alter the crystalline structure of CB, as already observed for HSAG [27] and CNT [28]. These results confirm that the pyrrole methodology for the functionalization of the sp^2^ carbon allotropes modifies the surface properties without affecting the crystalline structure of CB. Traditional oxidation methods of CB, for example, with HNO_3_, were shown to be able to destroy the CB structure, leading to graphene layers [41].

### 3.2. Preparation and Characterization of Rubber Composites

Two studies were performed, changing the rubber matrix and the extent of SP in the CB/SP adduct.

*Composites based on BR/NR.* In the first study, composites were based on BR and NR, and the hybrid filler system was made up of silica and CB (35 phr and 30 phr, respectively). As mentioned in the Introduction, this type of composite is suitable for a tire sidewall compound with a low hysteresis. CB/SP-6 was used in the partial replacement of silica: 66% by volume. Recipes are in Table 1. The same amount of the coupling agent, the silane TESPT, was used in both the composites. The compounding procedure is in Figure 2. In the first step, silica was mixed with the rubber in the presence of silane TESPT to allow the interaction of TESPT with silica and the occurrence of the silanization reaction, at least to some extent. CB/SP and CB were fed in the second step.

The crosslinking was performed with a sulfur-based system. The rheometric curves are in Figure 5 and data of minimum modulus M_L_, maximum modulus M_H_, induction crosslinking time t_s1_, optimum crosslinking time t_90_ and curing rate are in Appendix A.

Similar values of M_L_ were obtained. M_L_ is an index of the viscosity of the composite and, for composites with the same rubber matrix and loaded with the same volume of reinforcing fillers, is essentially due to the extent of the filler network, which appears, thus, to be similar in the two composites. A remarkably higher M_H_ was obtained with CB/SP. The values of M_H_ are due to the extent of both the crosslinking network and the filler network. In fact, the maximum amplitude explored through the shear test is not enough to completely disrupt the filler network. Values of induction and optimum times of vulcanization are similar for the two composites, whereas a remarkably higher curing rate was obtained for the composite with CB/SP. In the presence of similar M_L_, and taking also in consideration that the two composites contained the same amount of sulfur atoms, the higher value of M_H_ for the composite with CB/SP could be due to a more efficient vulcanization, with the formation of a higher crosslinking network density and/or shorter sulfur bridges. The higher M_H_ and curing rate could be due to the presence of sp^3^ nitrogen atoms on the CB surface. This hypothesis is proposed assuming that TESPT plays its role for the functionalization of the filler(s) rather than as sulfur donor. The silanization of CB/SP with a silane is discussed below in the text. However, further studies are needed to clarify this point.

Dynamic-mechanical properties were determined in the shear mode through performing strain sweep experiments in the range from 0.2% to 25%, as described in the experimental section. Data are in Table 4, and curves of G′ and tan delta are in Appendix A.

With respect to the reference composite, CB/SP led to lower values of ΔG′ and ΔG′/G′_γmin_, whereas the G″_max_ and the Tan δ_max_ are in line. The replacement of silica with CB/SP appears to reduce the Payne effect [42,43,44,45] and hence reduce the filler networking phenomenon.

Dynamic-mechanical properties were also determined in the axial mode through applying sinusoidal stress as described in the experimental section. Data of E′, E″ and Tan delta, measured at 10 °C, 23 °C and 70 °C, are in Table 5, and the dependence of E′ and Tan delta on temperature are in Figure 6, Figure 6a and Figure 6b, respectively.

With respect to the reference composite, CB/SP-6 led to remarkably higher values of E’, at every temperature, and lower values of Tan delta, particularly at high temperature. These results indicate a lower extent of the filler network and are in line with those from the shear tests.

Tensile properties were determined through quasi-static measurements. Data of stresses at different elongations and stress, strain and energy at break are in Table 6, and curves are in Appendix A.

CB/SP led to remarkably higher values of stress at every strain and to lower stress, elongation and energy at break. These results appear to be in line with the higher values of dynamic rigidity. To account for the lower values of the ultimate properties, though for both the composites remarkable values were obtained, the ability of CB/SP to interact with silica and to give potentially rise to filler agglomerates should be investigated, in future studies. Moreover, the viscosity of the composite could play a role.

*Composites based on IR.* In the second study, composites were based on IR as the only rubber and, analogously to the first study, a hybrid silica/CB filler system (35 phr and 30 phr, respectively) was used. As reported in the Introduction, this type of composite, with NR as the rubber, is suitable for internal tire compounds with low hysteresis. The recipes of the composites are in Table 2. The replacement of silica with CB/SP was performed at the same level as in the first study (66% by volume), and the two samples of CB/SP adducts, CB/SP-4 and CB/SP-5, were used. The main objective of this study was to investigate the effect of CB/SP adducts with a lower level of SP. Composites were prepared via melt blending with the same procedure used for the first study, shown in Figure 2.

Sulfur-based crosslinking was investigated with a rheometric test. Rheometric curves are in Figure 7, and data of M_L_ and M_H_, t_90_, t_s1_ and curing rate are in Appendix A.

The crosslinking curve of the composite with the whole amount of silica reveals the reversion phenomenon, absent for the composites with CB/SP, which also give rise to lower M_L_ values, particularly with CB/SP-4. The highest value of M_H_ was obtained with CB/SP-5. These results indicate that even a subtle difference of the SP amount in the CB/SP adduct have an appreciable effect on the crosslinking behavior and suggest that a higher amount of SP leads to higher composite’s viscosity and, particularly, to more efficient vulcanization. Moreover, they appear to be in line with those obtained in the first study.

Strain sweep experiments in the shear mode were carried out, as described in the experimental section, similar to the first study. Data collected from the experiments on both uncured and cured samples are considered. The curves of G′ vs. strain amplitude for the uncured and cured samples of Table 2 are to be seen in Figure 8a and Figure 8b, respectively, and the curves of Tan delta vs. strain amplitude for the uncured and cured samples of Table 2 are shown in Figure 9a and Figure 9b, respectively. Data of G′_0.2%_, G′_25%_, ΔG′/G′_0.2%_, G″_max_ and Tan(δ)_max_ for the uncured and cured samples are in Appendix A and in Appendix A, respectively, in the Appendix A. The curves of G′ for the uncured samples with CB/SP in Figure 8a overlap and lie below the curve of the reference silica sample, throughout the range of strain amplitudes. The lower values of ΔG′/G′_0.2%_ for the CB/SP samples suggest that the replacement of silica with the functionalized CB leads to a lower Payne effect [41,42,43,44]. The dependence of G′ on the strain amplitude for cured CB/SP-4 and CB/SP-5 samples appears to be different, as shown in Figure 8b. Higher values of G′ were obtained with CB/SP-5, and a crossover of the curves due to the CB/SP-5 and the reference silica samples can be observed: higher values of G′ at high strain amplitude were obtained with the CB/SP-5 sample. The replacement of silica with CB/SP leads to the reduction of ΔG′ and ΔG′/G′, and, hence, of the Payne Effect, as well as to the reduction of Tan delta. The reduction of ΔG′/G′, in relation to the reference composite without CB/SP, can be correlated with the amount of SP in the adduct. It is worth observing that these results are in line with those obtained in the first study and that even a slight difference in the SP content in the CB/SP adduct leads to an appreciable different behavior in the strain sweep test. Although it is challenging to compare data from different studies, based on different rubber matrices, the numbers in Table 4 and Appendix A allow for the calculation of a percentage reduction in ΔG′/G′, of approximatively 5, 8 and 12% for composites with CB-4, CB-5 and CB-6, respectively, compared to the reference composites without CB/SP. This elaboration should be intended only as a qualitative indication of the correlation between the extent of CB functionalization with SP and the modification of the properties of the reference composite.

Axial dynamic-mechanical properties were obtained as outlined in the experimental section. Data of E′, E″ and tan δ, measured at 10 °C, 23 °C and 70 °C, are in Table 7, and the dependence on temperature of E′ and Tan delta is in Appendix A.

The replacement of silica with the CB/SP adducts led to higher dynamic rigidity: higher values were obtained with higher SP content. As observed for the dynamic-mechanical properties in the shear mode, the values of E′ from axial measurements can also be correlated with the amount of SP in the CB/SP adducts. As already commented, it is indeed hard to compare values obtained in different studies with different polymer matrices. However, it can be noticed that the increase of E′, with respect to the reference composite without CB/SP, was (%) 8, 13 and 35 at 10 °C and 3, 11 and 34 at 70 °C for CB/SP-4, CB/SP-5 and CB/SP-6, respectively. These findings indicate the prevailing effect of CB/SP on the dynamic-mechanical properties of the rubber composite, in spite of the different rubber matrices and the different grades (different providers) of CB. The reduction of hysteresis was achieved with the greater amount of SP.

Quasi-static measurements were used to determine the composites’ tensile characteristics. The tensile curves are in Figure 10 and data are reported in Appendix A.

Higher values of stresses at every elongation and at break were obtained with CB/SP in the composites in place of silica. The highest values were for the sample with the highest amount of SP in the adduct. This larger mechanical reinforcement was accompanied by a lower elongation at break, though all the composites achieved high elongations. These findings are in line with what was observed in the first study. The ultimate properties appear to be closer to those of the reference composite in this second study. The higher viscosity of the rubber matrix could play a role. However, the ability of CB/SP to interact with silica and potentially to give rise to filler agglomerates could lead to lower elongation at break and it should be investigated in future studies. It is worth adding that data from optical microscopy revealed that the CB dispersion was not appreciably modified through the use of CB/SP in place of silica. Values are in Appendix A.

As mentioned in the Introduction, the replacement of silica with CB/SP could have a positive effect on the electrical conductivity of the elastomer composite. In this work, the electrical resistance was measured, as reported in the experimental section, for the samples of composites in Table 2. In particular, the data reported in Table 8 refer to the electrical properties of the silica and the CB/SP 4 compounds.

The measures show that a substitution of 66% in volume of silica with CB/SP 4 decrease the resistance by three orders of magnitude with respect to the reference.

### 3.3. Silanization of CB/SP

Preliminary studies have been conducted to investigate the reactivity of CB/SP with a sulfur-based silane such as bis(triethoxysilylpropyl)tetrasulfide (TESPD). Details are in the experimental section. In brief, headspace (HS) analyses were performed at 150 °C for 20 min, measuring the ethanol emission from the reaction of TESPD with CB, SP and CB/SP. In these reactions, the amount of TESPD and the molar TESPD/OH ratio were kept constant. The area of the ethanol GC peak was correlated with the extent of silanization [12]. In Figure 11, the chromatograms of the volatiles of the above-mentioned samples are reported and compared.

The area of the ethanol peak coming from the reaction of TESPD with SP is larger than the area of the pure TESPD. The area of the ethanol peak coming from the reaction of TESPD with CB/SP sample is the largest. These findings suggest that the OH of the serinol moiety of SP reacts with TESPD and that a silanization reaction of CB/SP can occur. The reactivity of the OH on the CB surface could explain the appreciable differences among the composites’ properties, observed even in the presence of a subtle difference in SP content in the CB/SP adducts.

### 3.4. On the Reactivity of CB/SP

The results discussed above show that CB/SP can replace silica in elastomeric composites, resulting in higher dynamic rigidity, lower hysteresis and higher stresses at every elongation in tensile tests. Moreover, it is worth highlighting the appreciable differences among composites containing CB/SP adducts with a subtle difference of SP content. These findings appear to support the working hypothesis reported in the Introduction: CB functionalized with SP acts as a polar filler and is able to interact with both silica and the coupling agent TESPT. The results of the silanization study discussed in the previous paragraph confirm the ability of CB/SP to react with sulfur-based silane. In a recent paper [46], some of the authors demonstrated the ability of serinol pyrrole to react with silica. SP was used as coupling agent of silica in place of TESPT. The reactivity with silica was due to the serinol moiety, which is on CB’s surface in the CB/SP adduct. Hence, it can be concluded that the OH on the CB surface can enhance the reactivity of CB with polymer chains, mediated either by silica or by the sulfur-based silane.

## 4. Conclusions

This work demonstrates that silica is not the sole nanostructured reinforcing filler for elastomer composites with low hysteresis. In fact, a furnace carbon black functionalized with SP was used in rubber composites based on diene elastomers and a CB/silica hybrid filler system, replacing 66% of silica by volume, thus obtaining more efficient vulcanization, a lower Payne Effect, higher dynamic rigidity, low hysteresis and increased stresses in tensile measurements. Thus, a furnace CB, even with relatively low surface area (80 m^2^/g), when functionalized with a polar modifier such as serinol pyrrole, can replace a significant portion of silica, leading to composites with even lower hysteresis. The functionalization of CB with SP is achieved through a simple mixing and heating process, yielding a high carbon efficiency without the need for catalysts and, potentially, for solvents. Traditional formulation and melt blending can be applied. The chemical modification of CB appears to play a major role. The OH groups facilitate interaction with silica and with silane TESPT and, thus, with the polymer chains. The sp^3^ nitrogen atom could be responsible for the efficient sulfur-based crosslinking and the high dynamic rigidity. The lower Payne effect and the lower hysteresis, compared to the silica-based composite, could be attributed to the synergy between chemical reactivity and the lower surface area of CB.

All the procedures here described can be easily scaled up to the industrial level. As mentioned in the introduction, a major player in the tire industry is currently undertaking this work.

This research highlights the potential of carbon fillers with polar groups in the development of elastomer composites with low environmental impact. Furnace CB is oil based and is not considered a sustainable material. CB from different sources are under development: bio char from renewable sources and circular char from end-of-life polymer products. These chars typically have low surface area and are not nanostructured. However, this work shows that these drawbacks could be overcome through chemical reactivity. Functionalization with pyrrole compounds appears to pave the way for the development of new, more sustainable generations of carbon materials.

## Figures and Tables

**Figure 1 polymers-15-03120-f001:**
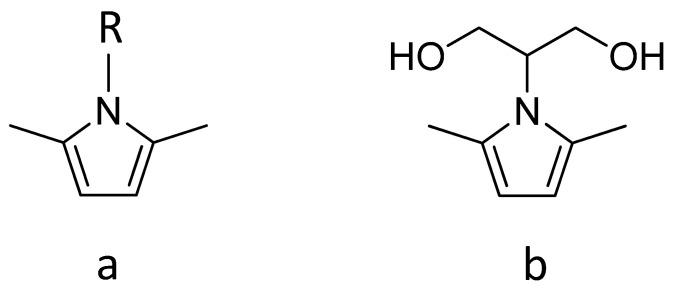
Pyrrole compounds. General chemical structure (**a**); 2-(2,5-dimethyl-1*H*-pyrrol-1-yl)-1,3-propanediol (serinol pyrrole, SP) (**b**).

**Figure 2 polymers-15-03120-f002:**
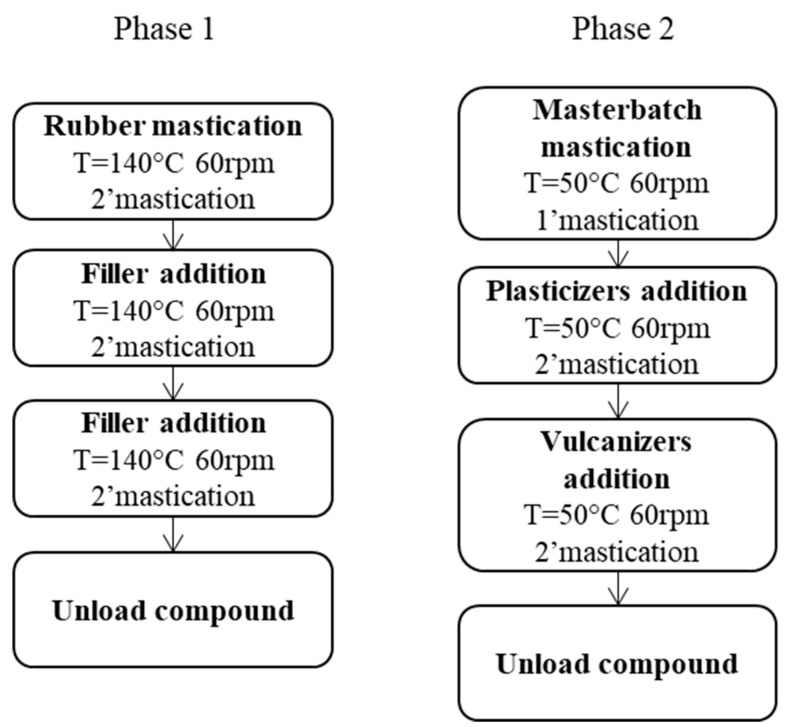
Block diagram of the process for the preparation of rubber composites.

**Figure 3 polymers-15-03120-f003:**
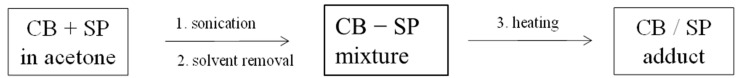
Procedure for the preparation of the CB/SP adduct.

**Figure 4 polymers-15-03120-f004:**
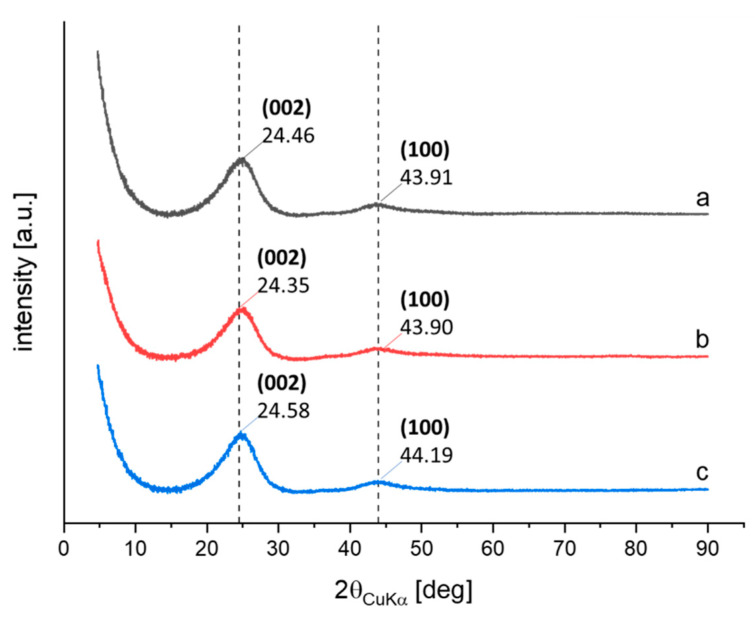
X-ray diffraction patterns of pristine CB N326 (**a**), CB/SP-4 (**b**), CB/SP-5 (**c**).

**Figure 5 polymers-15-03120-f005:**
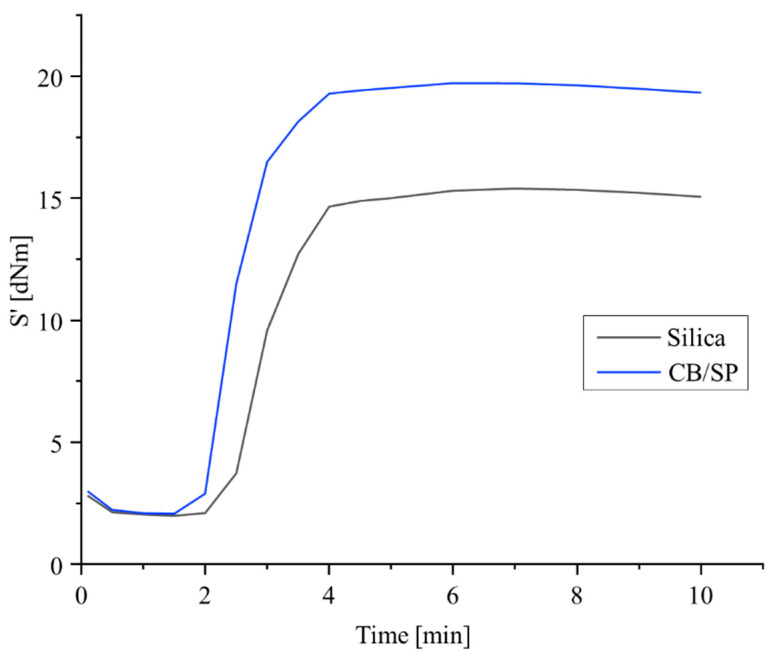
Rheometric curves for rubber composites of Table 1.

**Figure 6 polymers-15-03120-f006:**
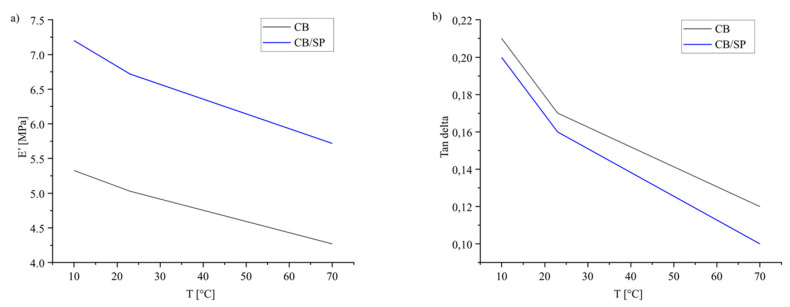
Dependence on temperature of E′ (**a**) and Tan delta (**b**) for the composites of Table 1.

**Figure 7 polymers-15-03120-f007:**
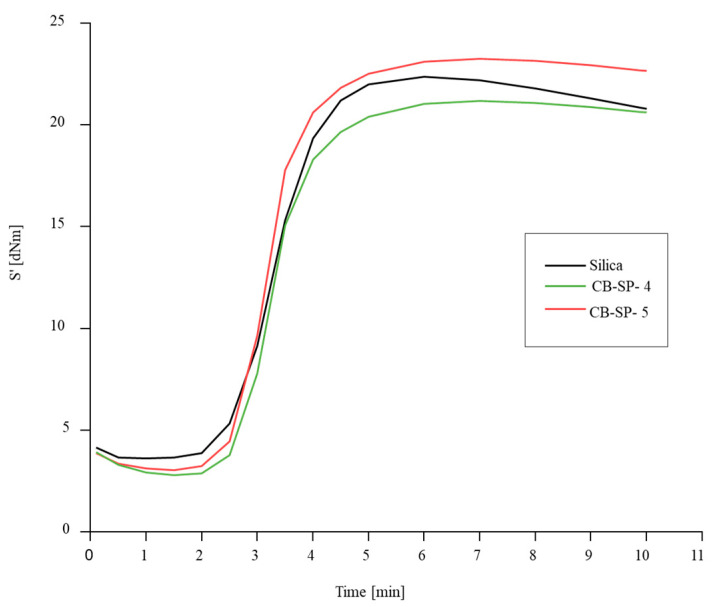
Vulcanization curves of IR-based compounds: torque versus time.

**Figure 8 polymers-15-03120-f008:**
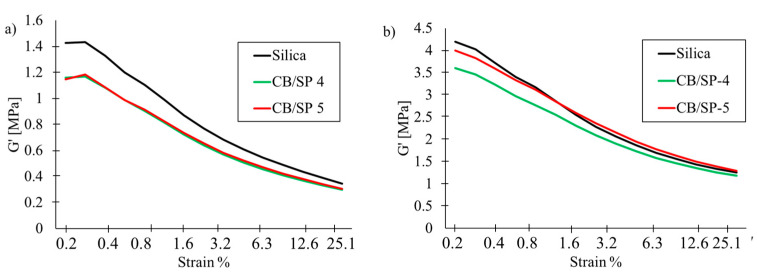
G′ vs. Strain curve for rubber composites of Table 2 (**a**) uncured (**b**) cured.

**Figure 9 polymers-15-03120-f009:**
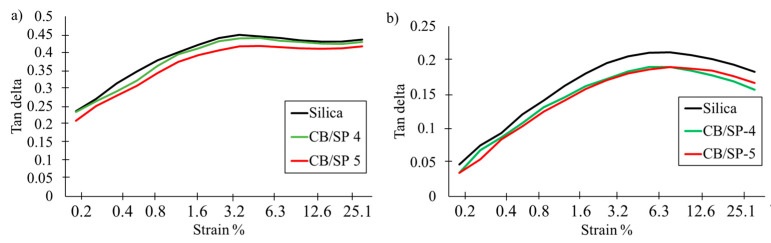
Tan delta vs. Strain curve for rubber composites of Table 2 (**a**) uncured (**b**) cured.

**Figure 10 polymers-15-03120-f010:**
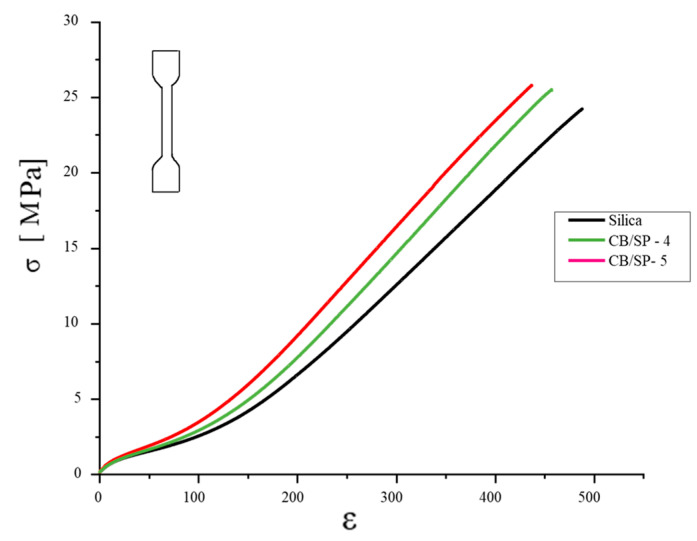
Tensile curves of composites of Table 2.

**Figure 11 polymers-15-03120-f011:**
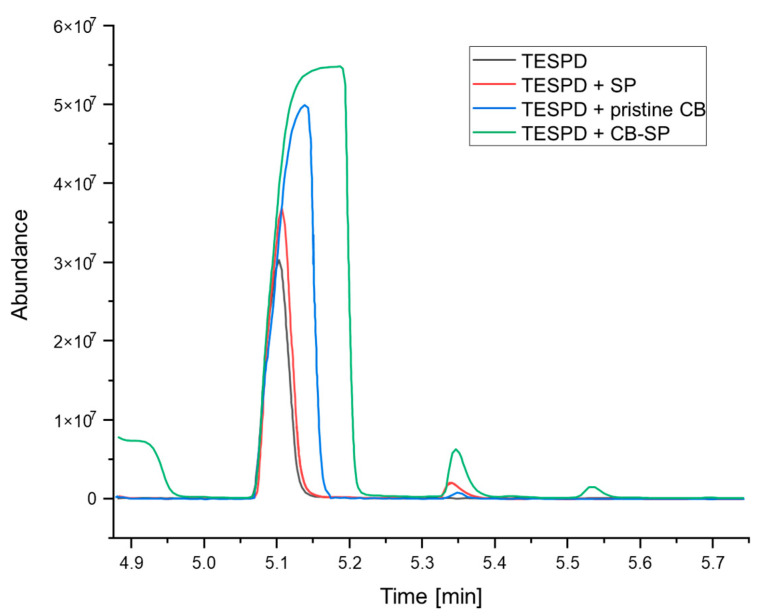
Chromatograms from HS-MS testing at 150 °C for 20 min of TESPD (black), TESPD + SP (red), TESPD + pristine CB (blue) and TESPD + CB/SP (green).

**Table 1 polymers-15-03120-t001:** Recipes of NR/BR-based composites with CB/SP-6 ^a^.

Ingredient	Silica	CB/SP-6
NR (SIR-20)	70	70
BR	30	30
Silane/CB	5.6	5.6
N326	30	30.0
Silica	35	12
CB/SP-6	0	19.72

^a^ Other ingredients (phr): stearic acid 2, ZnO 4, 6PPD 2, sulfur 2, TBBS 1.8, *N*-cyclohexylthiophthalimide 0.5.

**Table 2 polymers-15-03120-t002:** Recipe of IR-based composites with silica, CB N326 and CB/SP ^a^ as the fillers ^b^.

Ingredients	Silica	CB/SP ^b^
IR	100.00	100.00
Silica	35.00	12.00
CB/TESPT	5.60	5.60
CB N326	30.00	30.00
CB—SP	0.00	19.72

^a^ either CB/SP-5 or CB/SP-4. ^b^ Other ingredients (phr): stearic acid 2, ZnO 4, 6PPD 2, sulfur 2, TBBS 1.8, *N*-cyclohexylthiophthalimide 0.5.

**Table 3 polymers-15-03120-t003:** Mass losses of CB and CB/SP adducts from TGA analysis.

Sample	SP ^a^ (phc)	Reaction T (°C)	Mass Loss (%)			SP in CB/SP (phc)	F.Y. ^b^ (%)
			T < 150 °C	150 °C < T < 900 °C	T > 900 °C		
CB/SP-6	10	150	0.2	5.6	94.2	5.9	59
CB/SP-4	5	120	0.7	3.8	95.0 ^c^	4.0	80
CB/SP-5	10	120	0.4	4.8	94.1 ^d^	5.1	51

^a^ amount of SP in the reaction mixture; ^b^ functionalization yield (see Equation (1)); ^c^ residue: 0.6%; ^d^ residue: 0.7%.

**Table 4 polymers-15-03120-t004:** Dynamic-mechanical properties from shear tests of composites of Table 1.

	Silica	CB/SP-6
G′_0.2%_ [MPa]	2.52	2.30
G′_25%_ [MPa]	0.94	0.94
ΔG′ [MPa]	1.58	1.36
ΔG′/G′_0.2%_	0.67	0.59
G″_max_ [MPa]	0.15	0.15
Tan(δ)_max_	0.11	0.11

**Table 5 polymers-15-03120-t005:** Axial dynamic-mechanical properties of composites of Table 1.

	T (°C)	Silica	CB/SP-6
E’ (Mpa)	10	5.33	7.20
	23	5.03	6.72
	70	4.27	5.72
E’’ (MPa)	10	1.13	1.41
	23	0.86	1.04
	70	0.52	0.56
Tan δ	10	0.21	0.20
	23	0.17	0.16
	70	0.12	0.10

**Table 6 polymers-15-03120-t006:** Tensile properties of composites of Table 1.

	Silica	CB/SP-6
σ_100_ (Mpa)	2.21 ± 0.03	3.39 ± 0.03
σ_200_ (MPa)	6.28 ± 0.12	9.05 ± 0.11
σ_300_ (Mpa)	13.61 ± 0.25	17.58 ± 0.22
σ_300_/σ_100_	6.15 ± 0.22	5.18 ± 0.13
σ_B_ (Mpa)	30.34 ± 1.15	25.91 ± 1.19
ε_B_ (%)	504.22 ± 10.14	392.94 ± 7.15
Energy (J/cm^3^)	60.18 ± 2.54	41.16 ± 4.52

**Table 7 polymers-15-03120-t007:** Axial dynamic-mechanical properties of composites of Table 2.

	*T* (°C)	Silica	CB/SP-4	CB/SP-5
E′ [Mpa]	10	7.09	7.67	8.00
	23	6.51	6.91	7.22
	70	5.69	5.83	6.30
E″ [Mpa]	10	1.89	2.09	2.05
	23	1.49	1.64	1.60
	70	0.84	0.83	0.85
Tan (δ)	10	0.27	0.27	0.26
	23	0.23	0.24	0.22
	70	0.15	0.14	0.13

**Table 8 polymers-15-03120-t008:** Electrical resistance of composites of Table 2: reference silica and CB/SP-4 composites.

Sample	Resistance [MΩ]
Silica	1.9 ± 0.3
CB—SP 4	(5.4 ± 1.0) × 10^−3^

## Data Availability

Not applicable.

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
