# Peer review of "Adducts of Carbon Black with a Biosourced Janus Molecule for Elastomeric Composites with Lower Dissipation of Energy"

_polymers, 2023, doi:10.3390/polym15143120_

Round 1

Author Response

Dear Editor,

Here attached you may find the Answers to the Reviewers.

Let us thank them for the careful revision which allowed us to improve the quality of the manuscript.

We answered all the comments and questions.

We had the chance to add many experimental data and to add two co-authors.

Best Regards

Prof. M. Galimberti

Reviewer 2 Report

This manuscript reported an elastomeric composites with low dissipation energy by incorporating  serinol pyrrole-functionalized carbon black. However, there are some concerns and suggestions which, if addressed, would enhance the quality of the manuscript. These are provided below:

 1. Page 4, “Preparation of SP and CB/SP adducts”, the authors mentioned 1H NMR and 13C NMR, however, the reviewer did not find corresponding characterization?

2. Some important data including TGA analysis, DMA, should be provide in manuscript rather than in SI. And for TGA curves, the abscissa is more appropriate using temperature than time.

3. Figure 4 is very unclear, and it is recommended to improve the resolution of the picture.

4. The pictures of the whole manuscript and SI are very poorly prepared.

Based on above, although there is probably some valuable data, the manuscript can not be accepted in present version.

There are also some grammatical errors. Please proofread and correct. As for figure caption, it is recommended to illustrate the detailed composites other than using Table  X....

Author Response

(The authors gave the same response as above.)

Reviewer 3 Report

The article "Adducts of Carbon Black with a Biosourced Janus Molecule for Elastomeric Composites with Lower Dissipation of Energy" discussed interesting work. However, the following comments are recommended:

1-  Although the abstract was written in good, I always recommend including the numerical results at the end of the abstract to prove the improvement achieved in the work. 

2- Figure 1 is better if it is discussed in the materials section.

3- The contribution of the study at the end of the introduction should be improved /rewritten.

4- double check of the figure number. 

5- I prefer if an infographic of the process for the preparation of rubber composites is designed. 

6- Please insert the sample view used in the Tensile test.

7. The conclusion should be free from references.

8. Also, the brief result of the study should be discussed in the conclusion.

9. Some of the references are old. Can they replace it with the new one (if available)? 

Minor editing of English language required

Author Response

(The authors gave the same response as above.)

Round 2

Reviewer 2 Report

The authors have done careful revising according to the comments, I suggest it can be accepted.

The authors have done careful revising according to the comments, I suggest it can be accepted.